# Numerical Simulation of Subdam Settlement in Ash Disposal Based on CGSW Optimization

Hao Wang [1,*], Yong Wu [1], Yun Tian [1], Xuefeng Li [2], Zongyao Yang [3,*] and Lindong He [1]

1 College of Environment and Civil Engineering, Chengdu University of Technology, Chengdu 610059, China; ywu@cdut.edu.cn (Y.W.); 74616.cool@163.com (Y.T.); hole101112@163.com (L.H.)
2 Southwest Electric Power Design Institute Co., Ltd. of CPECG, Chengdu 610059, China; lixuefeng@swepdi.com
3 Chengdu Surveying Geotechnical Research Institute Co., Ltd. of MCC, Chengdu 610059, China
* Correspondence: wonghao16@gmail.com (H.W.); yangzy91@163.com (Z.Y.)

**Abstract:** The stacking of impermissible materials in the disposal of dry fly ash is unprecedented in the last 40 years of power plant management in China, and their effect on the stability of the whole facility is uncertain. Due to the lack of relevant treatment experience, a more comprehensive method such as numerical modeling must be adopted for the final design. This paper set up a borehole database from geological logging data to obtain the distribution of the coal gangue solid waste. Then, it established an accurate three-dimensional mesh model through Rhino. Based on elastic–plastic mechanics, the finite difference code Flac3D 6.0 was employed to study the risk of the coal gangue as a dam foundation. A comparative analysis of the influence of the displacement method and the composite foundation method on subdam deformation and differential subsidence was conducted. The simulation revealed that the composite foundation method showed the best reductions: 70.57% in shear failure, 97.83% in tension failure, and 22.63% in maximum subsidence. Ultimately, the optimum stone column diameter of 0.5 m and the spacing of 6 m were proposed due to the standard deviation.

**Keywords:** coal gangue solid waste; dry ash disposal; stone column; fluid–solid coupling; hydrodynamics

## 1. Introduction

China is the world's largest coal producer and consumer, and the proportion of China's coal production and consumption has risen to 50.69% and 54.33% of global capacity, respectively [1]. Over half of the coal consumption is applied to power generation. Fossil-fired power plants provide 60.80% of China's electricity capacity [2]. Significant quantities of mine tailing materials and post-production ash are generated in the processes of coal fire production and are mainly disposed of at surface ash disposal facilities near the power plants [3]. Leaching tests reveal that fly ash can have an extensive environmental impact and that it commonly releases heavy metals like arsenic, copper, cadmium, manganese, zinc, lead, and nickel into the surrounding ecosystem [4–7].

Surface ash disposal facilities are mainly composed of ash retention dams, drainage and flood discharge systems, and flood control dams. Ash retention dams are basically classified as retention dams or raised dams according to their construction methods. The height of retention dams remains constant once disposal begins, whereas raised dams are raised by the construction of a new subdam on the surface of under-consolidated tailing materials. Raised dams can be built by upstream, downstream, or centerline methods [8–10]. With the enhancement of the government's environmental awareness, ensuring the stability of fly ash disposal is increasingly crucial.

There are two main methods for ash disposal. The first one is the wet ash disposal method, in which ash is mixed with proportional water and pumped as slurry to the ash pond. In the second one, the dry ash disposal method, the fly ash is transported to the

construction industry, such as that for brick making or cement making, or it is carried by trucks to a dry ash disposal facility for temporary or permanent storage.

The No.4 dry ash disposal (DAD4) is the fourth and latest surface ash disposal facility of the Panjiang power plant; the facility is at a distance of 3.5 km from the main power plant. DAD4 is located in the province of Guizhou in southwestern China. This facility was constructed in phases based on the need for additional volume by the upstream method in 2014. One initial dam and fourteen subdams were designed with a final storage capacity of 26.30 million $m^3$ (Figure 1). The project design documents show that the initial dam was a rockfill dam with a height of 28 m and a crest axis length of 78 m, whereas the subsequent subdams were constructed with rolled fly ash, with crest lengths ranging from 124 m to 780 m, as illustrated in Figure 1. The initial dam building used a slope ratio of 1:2 on the upstream side and a slope ratio of 1:3 on the downstream side. The subdam building, however, used a slope ratio of 1:3 on the upstream side and a slope ratio of 1:4 on the downstream side. A geotextile was applied as an inverted filter on the inside slope while the outside slope was protected by a geotextile and a drystone wall. Ultimately, the crest of the last subdam will reach an elevation of 1720 m when the whole disposal area is fully filled. The crest of the latest subdam (subdam-VII) had reached an elevation of 1640 m by the end of September 2021, with a fill volume of 9.24 million $m^3$.

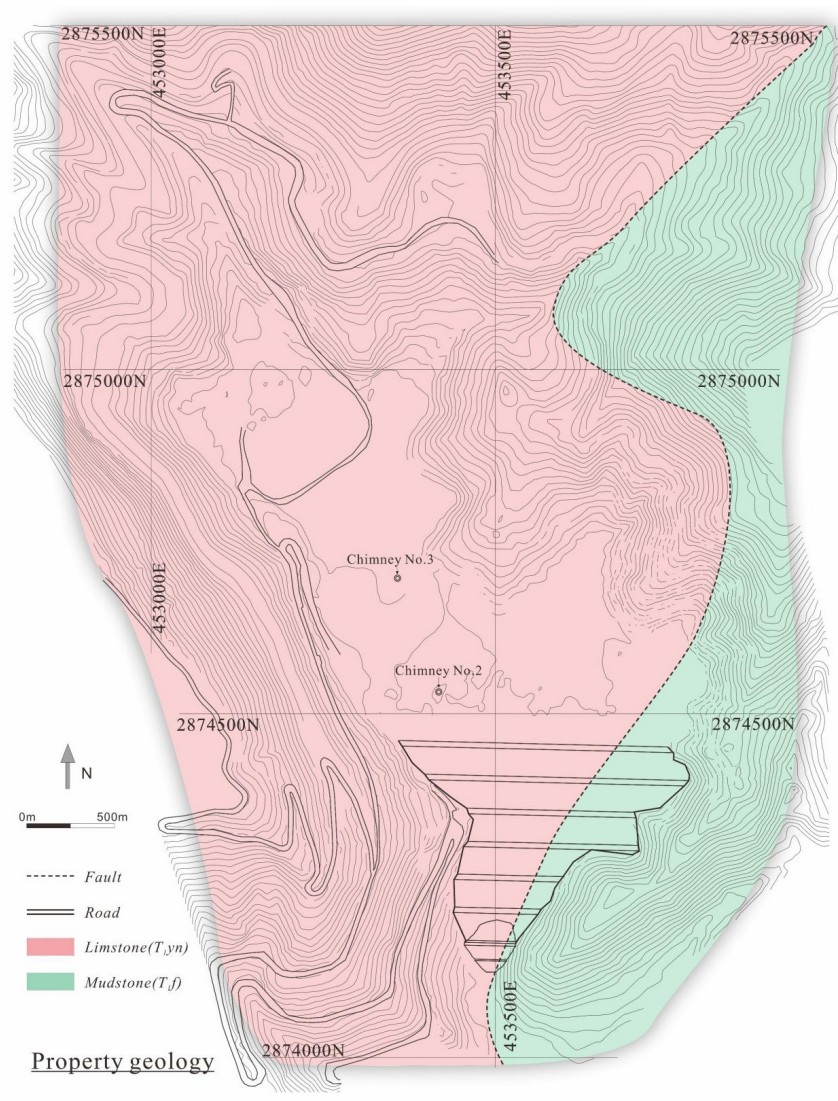

**Figure 1.** Geological property mapping of studied area.

Coal gangue solid waste (CGSW) is an inevitable solid waste formed by syngenetic sedimentation with a coal seam; it is produced in tunneling/mining/washing processes, with an emission amount of approximately 10–15% of domestic coal production [11]. The accumulated amount of piled-up CGSWs is about 6 billion tons so far; subsequently, it is anticipated to increase by 10–15% of annual coal production. The improper management of CGSW disposal could generate persistent air, agricultural soil, and groundwater pollution due to its content of heavy metal elements, thereby putting the nearby residents' health at risk [12–15]. As a governing agency, the National Energy Administration is supposed to guide relevant academic research institutions in CGSW reutilization by the building material industry, metallurgy industry, chemical industry, and light industry so that CGSW can have a positive environmental effect [16].

According to the design specification, only ash and desulfurization gypsum can be disposed in the dump area [17]. However, 121,186 m$^3$ of CGSW was improperly dumped between Chimney No.2 and Chimney No.3 by mistake (as shown in Figure 2) and then covered by fly ash. This buried CGSW is spatially located beneath the unbuilt subdam-XII, subdam-XIII, and subdam-XIV, where it faces a complex mechanical environment [18–21]. CGSW weathering produces clay minerals such as illite, kaolinite, and montmorillonite, and these clay minerals expand when they meet water. On the one hand, this expansion characteristic of clay minerals resulted in the blocking of the void channels inside the CGSW [22], which blocked the pores of underlying fly ash after migrating downward to the junction between the CGSW and fly ash through seepage flow; on the other hand, this formed an aquitard. Ignoring the CGSW would cause the uneven settlement of future subdams, leading to dam failure.

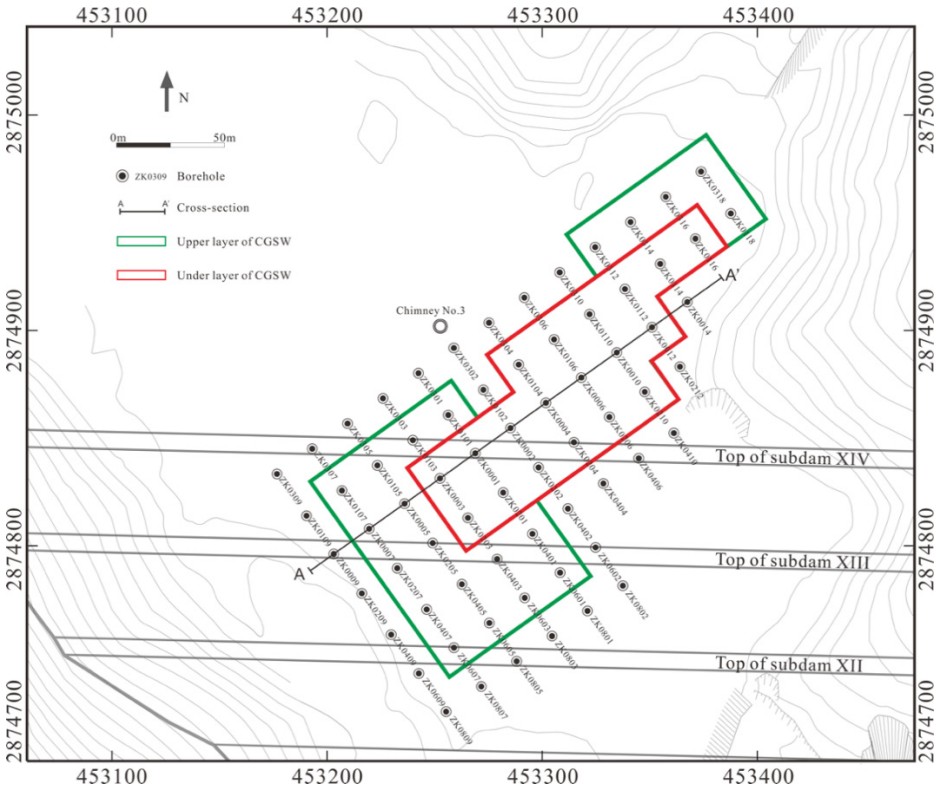

**Figure 2.** Plan view of CGSW and borehole distribution.

The fly ash is continually delivered to DAD4 from the power plant, which makes the buried CGSW go deeper. However, previous endeavors have not evaluated the effects of buried CGSW for subdam differential subsidence. Now, it is urgent to assess the hazard of the buried CGSW so that the management can preliminarily understand its impact on

the subdams, as well as on the safe operation of the entire ash dam, and can then make a proper decision.

Although the stability of the tailing dam in the valley context has been researched for more than 100 years, controlling the risk of a deep foundation to ensure dam stability is still challenging. Tailing dams collapse every few years. The international dam committee had recorded 366 global tailing dam failures spanning 1905–2020, and the rate of tailing dam failure is increasing [23–25]. The statistics show that upstream dams are much more likely to fail than centerline dams and downstream dams [26,27]. Generally, the reasons why these collapses occur can be summed up in one principal cause: a lack of a thorough comprehension of the key factors related to adequate design.

The finite difference method is one of the oldest methods for solving equation sets and differential equations. The FLAC finite difference program developed by ITASCA in the late 1980s has allowed this method to be widely used in geotechnical engineering numerical calculations. The FLAC uses the Lagrangian difference method, which, unlike finite elements, does not require the formation of an overall stiffness matrix; so, it is easy to correct for large deformations at each calculation time step. The coordinates and displacement increments are applied to the grids then move and deform along with the material. This overcomes the limitations of the traditional Eulerian method with relatively fixed meshes for material movement and deformation.

Other papers have reported the effect of mining on surface settlement [28,29] and have conducted stability analyses of tailing ponds [30] using Flac3D simulation. Sun Yulian et al. used static analysis with the Duncan tensor model to study the effect of different stages of subdam raising on the stress and deformation of ash dams [31]. Sitharam et al. calculated the dam safety values and failure probabilities from the kinetic point of view to analyze the stability of ash dams. Previous studies have analyzed the stability of dams from the seepage perspective. In this paper, rainfall and seismic factors are not considered for the time being. Firstly, an accurate CGSW distribution model is established based on the drilling database, and the effect of different CGSW treatments (the replacement method and gravel piles with different parameters) on the uneven settlement of the subdam is studied from the perspective of static analysis.

The previous studies on ash disposals are mainly divided into the categories of geotechnical parameters, the stability of ash dumps under seepage conditions, static stability, and dynamic stability. In this paper, an unprecedented challenge has been encountered: the localized concentration of CGSW in the ash dump area, which should not be present. Consequently, the treatment method has become a dilemma for the management. Therefore, this paper designs a series of orthogonal test scenarios to evaluate the impact of each scenario on the settlement of the proposed subdams and finally selects an optimal treatment option to provide recommendations based on the orthogonal test results. The previous studies basically focused on the bearing capacity [32–34], damage mode [35], deformation [36,37], and consolidation [38,39] of single piles or pile groups, and the depth of pile burial was less than 10 m. The study of the uneven settlement of the superstructure by deep pile groups has rarely been mentioned. In this paper, the pile groups are buried at a depth of about 80 m. Using a complex three-dimensional model established by numerical simulation, the effects of different pile group displacements on the settlement of the superstructures under complex original terrain conditions are calculated.

## 2. Geology and Distribution of CGSW

A case study using numerical modeling techniques was carried out to gain a better understanding of the controlling factors of dam foundation treatment. Data were provided by Panjiang Power Investment Co., LTD, Liupanshui City, China. This study is part of a feasibility study of the CGSW disposal concept, with the primary purpose of selecting an adequate stone column size and spacing through a design with numerical modeling. The relevant information about this dam is discussed in the following sections.

### 2.1. Geology

The geological property of the dam site is shown in Figure 1. The dam body is on top of Lower Triassic strata, which include limestone from the Yongningzhen Formation($T_1yn$) and mudstone from the Feixianguan Formation($T_1f$). In addition, the geological structure is very simple, with only one fault developed along the contact of the two strata.

### 2.2. Distribution Characteristics

Accurate distribution data of CGSW are the basis of subsidence prediction. We conducted an intensive geological engineering investigation to ascertain the CGSW's boundary and thickness. The shape of the buried CGSW was constrained by 68 boreholes. Drilling revealed that the buried CGSW covers an area of 20,000.00 m² with total volume of 125,780 m³ in an elevation range from 1631.67 m to 1643.78 m. The CGSW appears as two layers, both approximately parallel to the surface, as shown in Figures 2 and 3. The upper layer covers an area of 20,000 m², with a volume of 68,400 m³ and an average thickness of 3.42 m. The lower layer, by contrast, covers an area of 9500 m², with a volume of 57,380 m³ and an average thickness of 6.04 m.

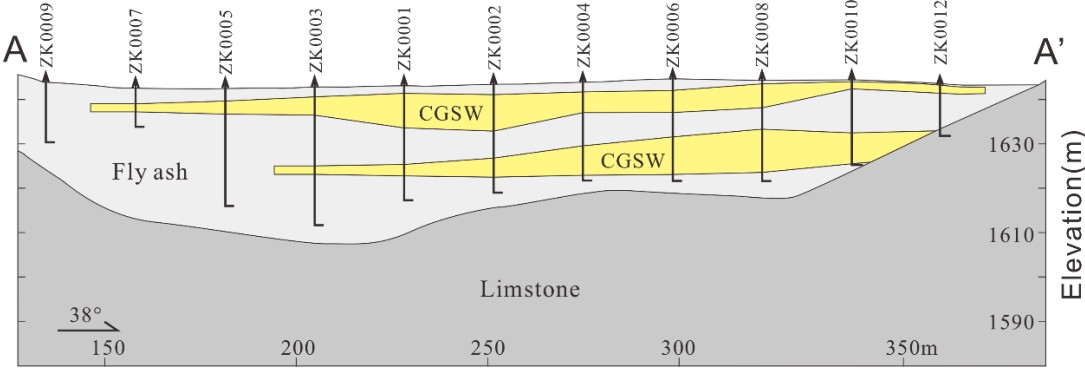

**Figure 3.** Simplified geological engineering cross-section.

## 3. Modeling and Test Scheme Design

### 3.1. Description of the Model

Before numerical simulation, the modeling process was completed on Rhino 6 (Robert McNeel Associates 2015), which is well known for three-dimensional modeling with powerful surface mesh building capabilities and excellent model structure generation, as shown in Figure 4. This software produces complex geometries and presents high compatibility with other file formats. Griddle 1.0 for Rhino 6, developed by Itasca Ltd., Minneapolis, MN, USA is a specific plugin to re-mesh the original surface mesh to build accurate 3D models. Surface meshes can be used as the boundaries of model meshes to generate high-quality tetrahedral or hexahedral meshes. The numerical model dimensions were approximately 400 m × 320 m × 200 m. It comprised rock mass, fly ash, two layers of CGSWs, and four subdams. The characteristics of each module are illustrated as follows:

1. Bedrock: The bedrock of DAD4 is composed of limestone and mudstone, derived from the Yongningzhen Formation and the Feixianguan Formation, respectively, and both are slightly weathered. The bedrock module was divided into 36,061 zones.

2. Fly ash: The fly ash was generated by a power plant 3 km away. It will reach a storage of 26.30 million m³ when DAD4 meets its ultimate capacity. In the Rhino model, this module was divided into 98,110 zones.

3. Coal gangue solid waste: The primary plan for the CGSW was for it to be dumped in DAD4 temporarily in the year 2016 without a further disposal strategy. So, it was buried eventually due to the continuous delivery of fly ash from the power plant. When they understood its possible impact on the stability of subdams and that it

might even cause dam failure, the management decided to carry out a geological engineering investigation in September 2021 to confirm the shape and distribution of the buried CGSW. The shape of the buried CGSW was constrained by 42 boreholes. The drilling revealed that the CGSW appeared in two horizontal layers, in parallel with each other and wrapped by fly ash. The mechanical parameters of the fly ash and bedrock were obtained through the laboratory geotechnical test, and the CGSW's mechanical parameters were from a large-scale triaxial test, which provided a good data basis for numerical simulation. The CGSW module was divided into 2887 zones.

4. Primary dam: The primary dam is an essential structure for surface ash disposal facilities constructed by the upstream method; as well as retaining fly ash, it also plays an important role in the drainage system to maintain a low phreatic surface. The primary dam in this case is a rolled rockfill dam made of waste limestone, with a slope ratio of 1:2 on the upstream side and a slope ratio of 1:3 on the downstream side. It rises up to the final height of 28 m, with a length of 121 m in axis; the volume of this starter dam is 97,168 m³. The primary dam module was divided into 31,960 zones.

5. Subdam: The subdam is mainly made up of compacted fly ash by vibratory rolling. The rolling test is carried out before construction to provide the best paving thickness, rolling times, optimum moisture content, and optimum compactness. As shown in Figure 4, the dimensions of these subdams are similar, with 70 m in bottom width, 8m in top width, 12 m in height; only the crest length varies from 124 m to 780 m. The subdam is entirely wrapped by an impervious geotextile, and the downstream slope is protected by dry rubble.

6. Stone column: The stone column is a mixture of breccia, gravel, and sand, with particle sizes of less than 100 mm. The uniformity coefficient (Cu) of the mixture must be greater than 5, with a curvature coefficient (Cc) between 1 and 3. All the column axes are 15 m to ensure the passing through of the coal gangue burial body. This module mesh must be less than one-fifth of the column diameter.

7. Dry ash disposal in the valley contains a perfect drainage system and flood-cutting system. Site investigation in 2021 found that the flood-cutting ditch and the drainage system, composed of a chimney, inclined shaft, and horizontal tube, were working well. The drilling operation did not reveal the water table, neither did the seven groundwater observation wells embedded in the early subdams. Under normal circumstances, the drainage system can discharge the infiltration rainwater within ten hours, retaining an unsaturated disposal facility for most of the time. The model was established under normal drainage circumstances; so, the influence of seepage on the uneven settlement was not considered in the simulation. Fast Lagrangian analysis of the continua in three dimensions (Flac3d), developed by Itasca Ltd., was used to provide recommendations for construction design.

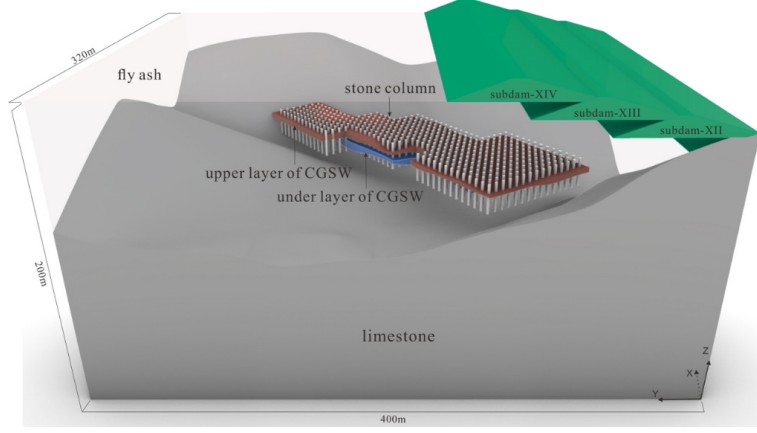

**Figure 4.** Three-dimensional model setup.

### 3.2. Constitutive Model and Boundary Condition

The Mohr–Coulomb (MC) criterion was chosen for all the modules (subdams, fly ash, CGSW, stone column, and bedrock) in the numerical model. The MC criterion was a simple elastic–plastic criterion that contains five parameters. These material parameters of the five different geological units were estimated from the laboratory geotechnical tests and are presented in Table 1.

**Table 1.** Static geotechnical properties of materials.

| Material | Density (kg/m$^3$) | Cohesion (KPa) | Friction (°) | Bulk Modulus (MPa) | Shear Modulus (MPa) | Tension (KPa) |
|---|---|---|---|---|---|---|
| Limestone | 2.69 | 650 | 40 | 11,000 | 11,000 | 6000 |
| Fly ash | 1.41 | 26 | 30 | 30 | 28 | 12 |
| CGSW (saturated) | 2 | 1 | 33 | 8.5 | 6.4 | 0 |
| CGSW (dry) | 1.65 | 26 | 30 | 1200 | 1200 | 0 |
| Stone column | 1.85 | 390 | 42 | 10,000 | 800 | 0 |
| Subdam | 1.53 | 30 | 23 | 32 | 30 | 14 |

The failure criterion used in this paper was a composite Mohr–Coulomb criterion with a tension cutoff, as illustrated in Figure 5. The magnitude relationship of the three principal stresses was present as $\sigma_1 \leq \sigma_2 \leq \sigma_3$. The failure envelope $f(\sigma_1, \sigma_3) = 0$ was defined from point A to B by the Mohr–Coulomb failure criterion $f^s = 0$ with

$$f^s = -\sigma_1 + \frac{1 + sin\varphi}{1 - sin\varphi}\sigma_3 - 2c\sqrt{\frac{1 + sin\varphi}{1 - sin\varphi}} \tag{1}$$

and from point B to C by a tension failure criterion of the form $f^t = 0$ with

$$f^t = \sigma_3 - \sigma^t \tag{2}$$

where $\varphi$ is the friction angle, $c$ is the cohesion, $\sigma^t$ is the tensile strength, and its maximum value of $\sigma^t$ is given by

$$\sigma^t_{max} = \frac{c}{tan\varphi} \tag{3}$$

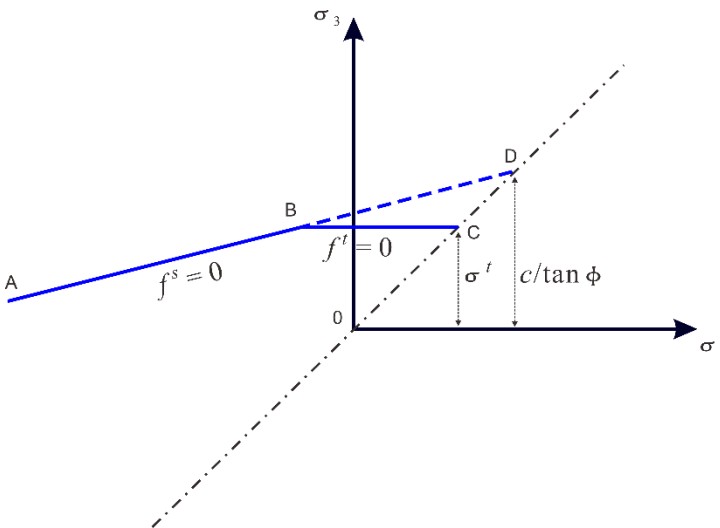

**Figure 5.** Mohr–Coulomb failure criterion in Flac3D.

The incremental expression of Hooke's law in terms of the generalized stress and stress increments has the form

$$\begin{cases} \Delta\sigma_1 = \alpha_1\Delta\varepsilon_1^e + \alpha_2\left(\Delta\varepsilon_2^e + \Delta\varepsilon_3^e\right) \\ \Delta\sigma_2 = \alpha_1\Delta\varepsilon_2^e + \alpha_2\left(\Delta\varepsilon_2^e + \Delta\varepsilon_3^e\right) \\ \Delta\sigma_3 = \alpha_1\Delta\varepsilon_3^e + \alpha_2\left(\Delta\varepsilon_2^e + \Delta\varepsilon_3^e\right) \end{cases} \tag{4}$$

where $\alpha_1$ and $\alpha_2$ are material constants defined in terms of the shear modulus, $G$, and bulk modulus, $K$, as

$$\begin{cases} \alpha_1 = K + \frac{4}{3}G \\ \alpha_2 = K - \frac{2}{3}G \end{cases} \tag{5}$$

For the boundary conditions, the free boundaries were applied to the top faces while the roller boundaries were applied to all the other sides of the model. The vertical velocity of the bottom and the horizontal velocity of the four vertical sides were constrained to a minimum to avoid unwanted stress concentrations. In-situ stress distribution was initialized by calculating after the application of gravity in the domain.

### 3.3. Design Basic

It was assumed that the settlement deformation and horizontal displacement of the dam body caused by geostatic stress had finished, indicating that the subsequent settlement deformation and horizontal displacement were caused by the new subdam. Accordingly, the initial stress of the bedrock was calculated first; then, we calculated the stress distribution generated by the primary dam and its corresponding fly ash. Each unit of the subdam with its corresponding fly ash was loaded and calculated successively to convergence until subdam-XIV was loaded. The CGSW is loaded at the same time as the subdam-VII and the corresponding fly ash.

Advanced numerical simulation by Flac3D 6.0 software, in conjunction with geotechnical investigations and laboratory geotechnical tests, can reveal the possible mechanisms of uneven settlement (subdam stability) caused by CGSW, providing a reference for selecting reasonable engineering disposal methods.

### 3.4. Test Scheme Design

The dynamic compaction method, displacement method, and composite foundation method are common methods of foundation treatment [40–43]. The dynamic compaction method compacts the gravel soil, silt, and miscellaneous fill soil foundation with dynamic compaction instruments. The replacement method can be used to excavate all CGSW and replace it with fly ash. The composite foundation method utilizes a backfill mixture of breccia, gravel, and sand in the process of vibration compaction to form columns [44–48]. This method requires consideration of the fact that different diameters and spacing of columns could produce different stresses on the soil between them, resulting in the production of different settlement resistance effects [49,50]. The management and investigation department prefers the composite foundation method; by filling the stone column to treat the gangue layer, the stone column can be not only conducive to the drainage of the upper stagnant water from the gangue body, it can also enhance the strength of the foundation.

To find an economical and rational construction scheme, ten construction schemes were designed for comparison, one of which was the excavation backfill technique; the rest were stone column construction techniques. The orthogonal test theory was used to design the stone column scheme. Three measurements of the stone column diameter were selected: 0.5 m, 1 m, and 1.5 m, and three measurements of the center distance were chosen: 3 m, 6 m, and 9 m. All the stone columns were 15 m in length, which was sufficient to pass through the CGSW.

In order to observe the settlement condition of the subdam, 79 monitoring points were set every 10 m, from the right shoulder to the left shoulder on the central axis of the subdam-XIV' crest (as illustrated in Figure 6—up). The displacement history of all the

monitoring points and the local unbalanced stress ratio of the whole model were recorded. When the value of the local unbalanced stress ratio was less than $10^{-5}$, the simulation ended, which indicated that the subdam settlement process had terminated. Therefore, the displacement of the monitoring points can represent the displacement of the corresponding position in the subdam crest.

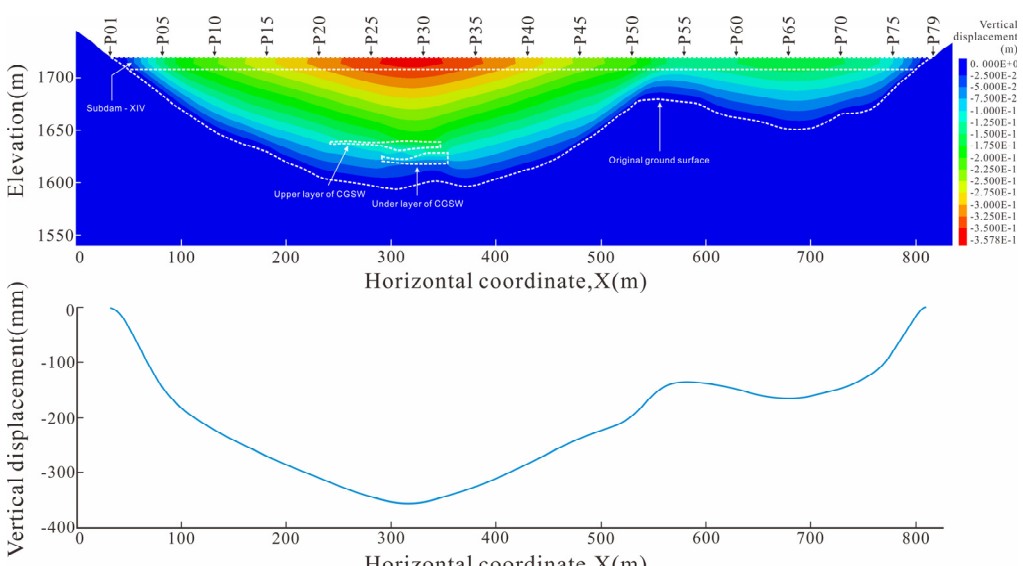

**Figure 6.** Up—numerically predicted section subsidence under original condition and location of monitoring points. Down—vertical displacement of top surface.

The subsidence curve, while providing a visual representation of the differences in settlement at each monitoring point on the top of subdam-XIV, did not provide an evaluation criterion for us to determine which scheme produced a better inhomogeneity of subsidence values. This paper introduces standard deviation (SD), a statistical indicator that reflects the amount of variation or dispersion of a set of values [51]. It is calculated as follows:

$$\sigma = \sqrt{\frac{\sum_{i=1}^{n} x_i - \overline{x}^2}{n-1}} \tag{6}$$

where $\sigma$ = standard deviation; $x_i$ = subsidence value of the $i$th monitoring point; $\overline{x}$ = average value of the parameter $x$; and $n$ = number of values of $x$ (number of monitoring points).

## 4. Results

The coal gangue here is mainly composed of mudstone, carbonaceous mudstone, and shale. Physical weathering combined with chemical weathering causes variation in the mineral formation and microstructure; this rubble will turn into a soft foundation in the foreseeable future and lead to subsidence on the ground surface [52]. According to the ash dam design scheme, this CGSW is situated 70 m apart from the topmost subdam in the vertical direction. Its declining bearing capacity may not be sufficient to prevent dam destruction under the increasingly uneven settlement.

In order to facilitate observation and description, we take here for analysis the section where the central axis of subdam-XIII is located. It can be seen that the section can be divided into two parts bound by the original topography; the lower part is hard limestone, and the upper part is composed of fly ash, CGSW, and subdam.

Once the calculations reached equilibrium, the maximum settlement value was obtained by counting the vertical displacement values of the 79 monitoring points on the central axis of the dam top. Using a loop code to traverse all the zones, the code "i = zone.state (z, bavg)" was used to identify the plastic state of all the zones in the model

and to count the volume of all the zones whose plastic state showed tensile failure and shear failure, respectively.

$V_i^t$, $V_i^s$ denote the total volume of the tension failure zones and shear failure zones in the $i$th scheme, respectively, and the total volume of the model is denoted by $V$. The percentage of shear failure zone $P_i^s = V_i^s/V$, and the percentage of tension failure zone $P_i^t = V_i^t/V$. $D_i^s$ is named as the reduction degree of the shear failure zone of the $i$th scheme and $D_i^t$ as the reduction degree of the tension failure zone of the $i$th scheme. They are computed by $D_i^t = (P_0^t - P_i^t)/P_0^t$ and $D_i^s = (P_0^s - P_i^s)/P_0^s$.

## 4.1. Deformation Analysis

Figures 7 and 8 are segmented into separate diagrams due to typography. The left part of these two figures (a0~a10) shows the displacement vector diagrams of the section under different schemes, while the right part (b0~b10) shows the plastic zone distribution of the section under these schemes. The series number of the subplots corresponds to the series number of the schemes in Table 2. This means a0 stands for the displacement vector of scheme 0 in Table 2; accordingly, b0 stands for the plastic zone distribution of scheme 0 in Table 2. As we can see from Table 2, scheme 0 is the original condition, and scheme 1 adopts the displacement method; however, schemes 2~10 use the composite foundation methods. Schemes 0~10 can be divided into three groups: schemes 2~4 adopt a column diameter of 0.5 m; schemes 5~7 adopt a column diameter of 1 m; and the column diameter in schemes 6~8 is 1.5 m. The column spacing in each group is, respectively, 3 m, 6 m, and 9 m.

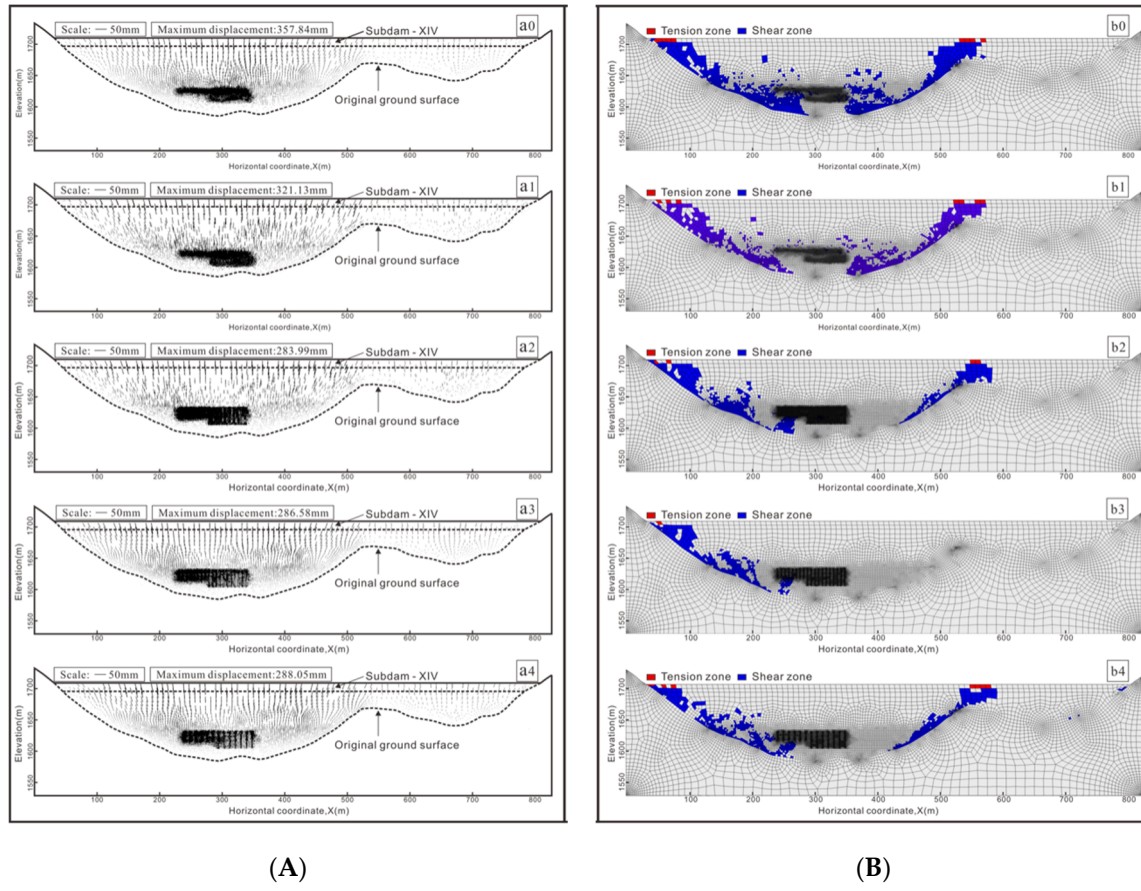

(**A**)                    (**B**)

**Figure 7.** Displacement vector (**A**) and plastic zone distribution (**B**) of the section under original condition, displacement method, and composite foundation method; the stone columns' diameters are 0.5 m.

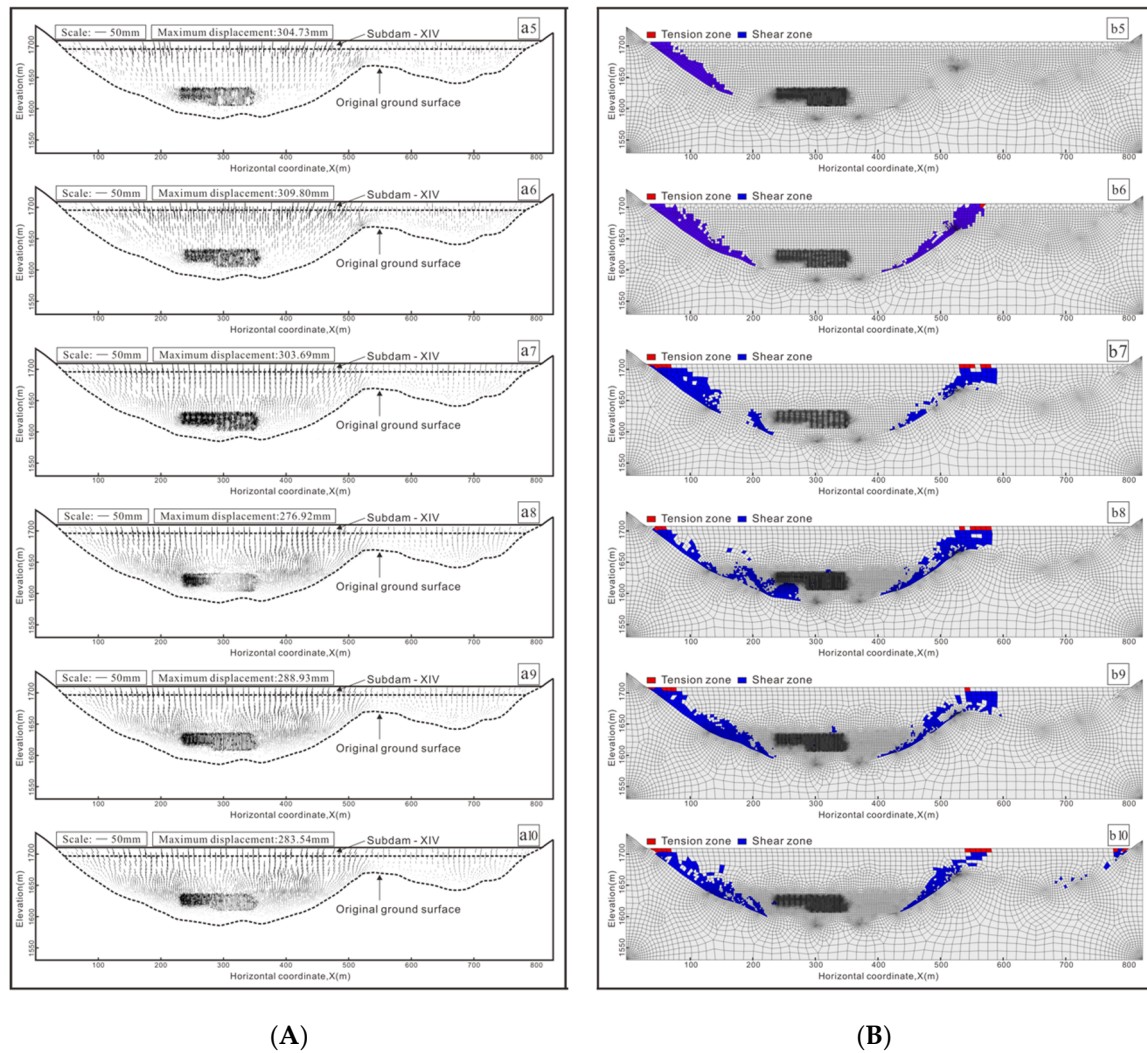

**Figure 8.** Displacement vector (**A**) and plastic zone distribution (**B**) of the section with stone columns whose diameters are 1 m and 1.5 m.

As shown in the displacement vector diagrams, each arrow represents the displacement state of a single grid point. The length and the direction of the arrow represent the displacement magnitude and direction of the grid point, respectively. Significantly, the quantity of arrows is only related to the meshing and is in proportion to the number of grid points. It is easy to see that the arrows exist mainly above the original ground surface—as shown in Figure 7A. In effect, this demonstrates that deformation occurs almost in the entire fly ash, CGSW, and subdam-XIV; in contrast, the bedrock hardly deforms.

The blue and red colors symbolize, respectively, the shear failure zones and tension failure zones, as shown in Figure 7B. The shear failure zones preferred to develop near the original ground surface in the fly ash module and CGSW module, while the tension failure zones only occurred in the subdam module; moreover, no plastic damage developed in the bedrock module. We can see that shear failure and tension failure could not be avoided, no matter what kind of method was used. But the composite foundation method was more effective as it reduced the failure zone volume of the section and improved the safety factor of the subdam (Figure 7). The stone column scheme with a 0.5 m diameter and 6 m center distance (as illustrated in Figure 7(b3)) reduced the shear failure zone by 60.61% and reduced the tension failure zone by 87.75%. In the meantime, the stone column scheme with a 1 m diameter and 3 m spacing (as illustrated in Figure 8(b5) reduced the shear failure zone by 70.57% and reduced the tension failure zone by 97.83%.

**Table 2.** Summary of orthogonal experiments and experimental results.

| Serial Numbers of Scheme | Parameter of Stone Column | | Subsidence Results | | Plastic Zone Information | | | |
|---|---|---|---|---|---|---|---|---|
| | Center Distances/m | Diameters/m | Maximum Subsidence/mm | Reduction Degree of MS | Percentage of SFZ | Percentage of TFZ | Reduction Degree of SFZ | Reduction Degree of TFZ |
| 0 | - | - | 358 mm | - | 3.60% | 0.16% | - | - |
| 1 | - | - | 320 mm | 10.615% | 2.68% | 0.13% | 25.63% | 18.93% |
| 2 | 3 m | 0.5 m | 283 mm | 20.950% | 2.80% | 0.08% | 22.24% | 50.62% |
| 3 | 6 m | 0.5 m | 285 mm | 20.391% | 1.42% | 0.02% | 60.51% | 87.75% |
| 4 | 9 m | 0.5 m | 287 mm | 19.832% | 2.79% | 0.12% | 22.41% | 24.72% |
| 5 | 3 m | 1 m | 304 mm | 15.084% | 1.06% | 0.00% | 70.57% | 97.83% |
| 6 | 6 m | 1 m | 309 mm | 13.687% | 2.85% | 0.06% | 20.85% | 60.03% |
| 7 | 9 m | 1 m | 302 mm | 19.642% | 2.62% | 0.15% | 27.15% | 3.89% |
| 8 | 3 m | 1.5 m | 277 mm | 22.626% | 3.01% | 0.13% | 16.40% | 20.06% |
| 9 | 6 m | 1.5 m | 288 mm | 19.553% | 2.61% | 0.08% | 27.41% | 49.55% |
| 10 | 9 m | 1.5 m | 283 mm | 20.950% | 2.11% | 0.21% | 41.27% | 28.96% |

Note: MS = maximum subsidence, SFZ = shear failure zone, TFZ = tension failure zone.

*4.2. Subsidence Analysis*

Figure 6 (up) showed the vertical displacement of the section when subdam-XIV was loaded. The dotted line sketches the original surface, fly ash, CGSW, and subdam-XIV. Vertical displacement on the top surface was increased as the thickness of the fly ash beneath increased. The maximum subsidence value was 0.358m near monitoring point P29. Figure 6 (down) plots the fitted curve of the settlement around all the monitoring points. It is noted that a double-valley shape appears (the CGSW is distributed in the left valley), resulting in a fitting curve that is similar to the original ground surface, which indicates a strong correlation between them. The analysis revealed that the displacement value of the monitoring point was mainly affected by the thickness of the fly ash below it. The greater the thickness of the fly ash, the larger the displacement value of the monitoring point.

The results of the predicted section subsidence of the 10 schemes when subdam-XIV was loaded are shown in Figure 9. Table 2 summarizes the maximum subsidence of subdam-XIV's top surface in the 10 schemes according to the MC failure criterion. It is remarkable that the series number of subplots in Figure 9 is consistent with that of the scheme in Table 2. The results indicate that all the construction schemes will reduce the subsidence value of subdam-XIV. Among them, the displacement method performed the worst with a maximum subsidence of 320 mm, a reduction of 10.615% compared to the original condition (scheme 1, subplot 1 of Figure 9). The composite foundation method performed the best; scheme 8 ended up with a maximum subsidence of 277 mm, a reduction of 22.625% (see Figure 9).

Figure 10 demonstrates the comparison of the fitted settlement curves of all the schemes. The numbering of the curves corresponds to the numbering of the schemes in Table 2. Among them, the solid black line (line 0) represents the settlement of the original condition, and the solid black line (line 1) represents the replacement method. Meanwhile, the colored line (line 2~10) charts the settlement of the composite foundation method, where the green line, blue line, and red line, respectively, express the stone column diameters of 0.5 m, 1 m, and 1.5 m; the solid, dotted, and dashed lines represent the column distances of 3m, 6m, and 9m, respectively.

These curves formed by the fitted settlement values of the monitoring points show almost no difference in the right valley segment (x > 560). A significant difference exists between the 0 m~560 m segment in the x-axis, where various sizes of stone columns are plugged. Thus, the variation of curves is mainly observed in this segment. The contrast of curve 0 and curve 1 demonstrates a similar variation tendency where the curves show a downward trend first and reach the maximum settlement at X = 320, then both turn upward. These two curves have different decline amplitudes; the peak of curve 1 is 10.61% less than

that of curve 0. The contrast of curve 0 and curves 2~10 indicates that both diameter and spacing can affect the uneven settlement of the subdam.

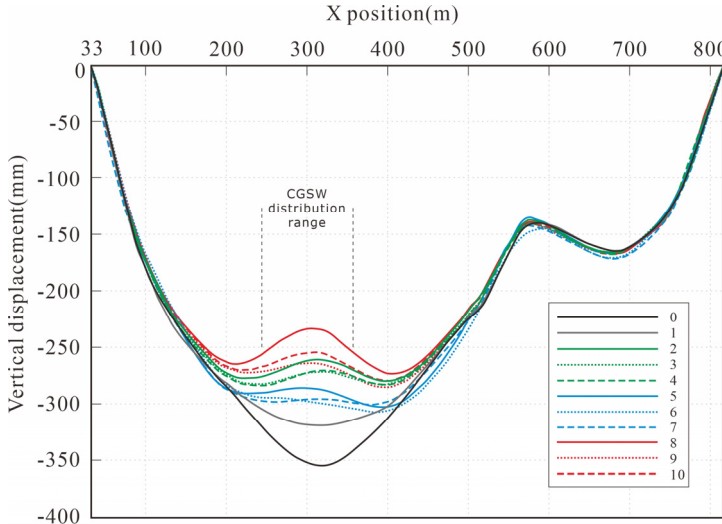

**Figure 9.** Comparison of numerically predicted section subsidence for 10 schemes.

**Figure 10.** Comparison of fitted subsidence curves for all schemes.

Curves 2~10 also decline first but then turn upward at X = 220 m and turn down again after forming a crest near X = 320 (except for curve 6, which just decreases in slope and peaks at X = 400 m), which generally presents a subordinated inverted double peak in the large valley area.

Both the diameter and the center distance of the stone column have an effect on the uneven settlement of subdam–XIV. According to curves 2, 5, and 8, which have the same stone column center distance but different diameters, we can tell that the maximum subsidence reduces with the increasing diameter when the diameter is 0.5 m. Curves 3, 6, and 9 (or curves 4, 7, and 10) indicate that the maximum subsidence increases and then decreases when the diameter is 1 m or 1.5 m (as also illustrated in Figure 11). When the center distances are the same, the maximum subsidence increases and then decreases with the increase in diameter.

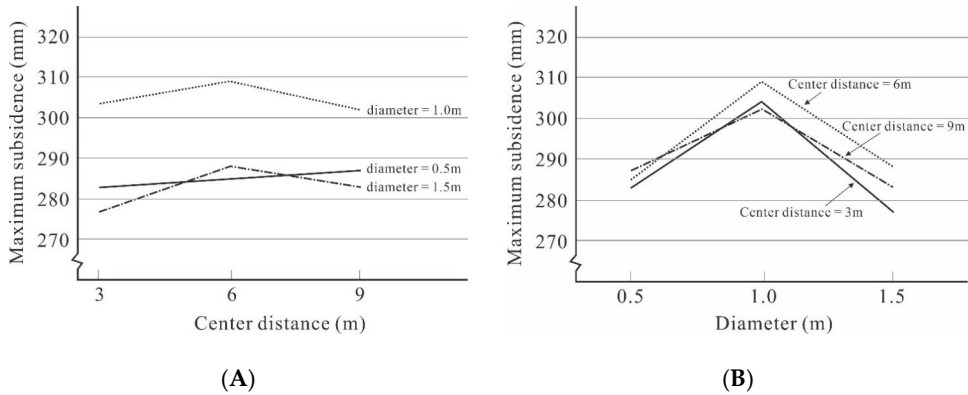

(**A**)                                (**B**)

**Figure 11.** Variation of maximum subsidence under same diameter (**A**) and variation of maximum subsidence under same center distance (**B**).

*4.3. Standard Deviation Analysis*

We know that the smaller the SD of a set of data, the less discrete the dataset is, indicating less inhomogeneity of the settlement. For each scheme in Table 2, the settlement values of 79 monitoring points form a set of data samples. So, each data sample produces an SD, resulting in a final total of 11 SDs. The SD of the settlement values for each scheme was placed into a bar chart (Figure 12) to facilitate our better observation. It can be seen that scheme 0 (original working condition) produces the largest SD of 96.24 mm. Schemes 1, 5, 6, and 7 produce SDs of 80 to 90 mm, and Schemes 2, 3, 4, 8, 9, and 10 produce SDs of 70 to 80 mm.

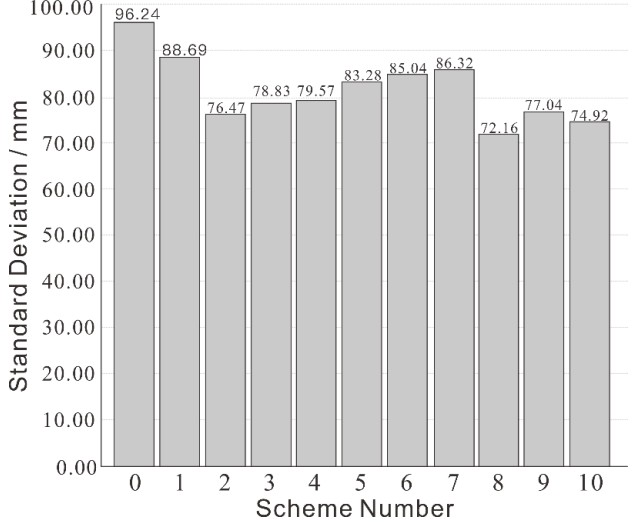

**Figure 12.** Standard deviation histogram.

*4.4. Analysis Summary*

Choosing a suitable construction scheme requires the consideration of various factors. Here, we give priority to the scheme that can minimize the profile deformation. As seen in Figures 6 and 7, scheme 3 and scheme 5 can effectively reduce the dam deformation and eliminate the tension damage zone in the middle of the subdam; so, these two options are preferred. According to the SD analysis, the SD of scheme 3 is 78.83, and the SD of scheme 5 is 83.28. This indicates that scheme 3 produces lower settlement inhomogeneity of the subdam; so, scheme 3 is recommended as the final construction scheme.

## 5. Conclusions

Although dry ash disposal in valley has been used and studied for more than 40 years, the soft foundation problems of the subdam are rarely encountered. With single dam foundation calculation, it is hard to evaluate the consequences of uneven settlement. Carrying out an effective and economical foundation treatment method that ensures dam stability is challenging. The primary goal of this study was to verify CGSW distribution by drilling engineering and then to initially evaluate the influence of CGSW on the proposed subdam. The orthogonal test theory was utilized for the development of various construction strategies for the possible soft foundation using numerical simulation. The finite difference code FLAC3D 6.0 was then employed to carry out stone column studies on the subdam stability to examine the effect of center distance and diameter. Based on the modeling results, the optimal construction scheme was selected after deformation and subsidence analysis. The key observations from the scheme assessment are:

1.  CGSW distribution based on borehole data can help to build a more accurate FLAC3D digital model; this is the basis of the deformation analysis and differential settlement analysis under various subsequent construction schemes.

2.  If deformation or differential subsidence on top of the CGSW cannot be effectively controlled, it could cause damage to the subdam, such as shear failure or cracking. The subdam stability is dominated by the differential subsidence rather than the absolute magnitude of subsidence. The modeling results indicate that the subdam differential settlement caused by CGSW weathering may threaten the stability of the overlapping subdam-XIV, leading to tension failure in the right and middle abutment.

3.  Both the replacement method and the stone column methods helped to reduce the subdam subsidence value. The displacement method could provide only a 10.615% reduction in the maximum subsidence value; the stone column methods worked better. Scheme 2, 3, 8, and 9 produced reductions of over 20%; scheme 8 provided the best reduction of 22.391%. The deformation analysis showed that scheme 3 and scheme 5 had the best effect in that they both eliminated the tension failure zone in the middle of subdam-XIV. The shear failure zone and tension failure zone in scheme 3 were reduced by 60.51% and 87.75%, respectively, compared to the original condition (scheme 0). Scheme 5 reduced the shear failure zone by 70.75%; meanwhile, it reduced the tension failure zone by 97.83%.

4.  Generally, it is known that under longitudinal loading conditions three types of deformation or damage, known as "bulge deformation", "shear damage", and "block damage", may occur in the gravel pile. The diameter and spacing of the gravel pile both affect the settlement of the superstructure within the diameter range of 0.5–1.5. Regardless of the pile spacing value, the maximum settlement value of the top surface always exhibits an ascending trend followed by a descending trend. This may be due to the fact that as the diameter (plane replacement rate) increases, the resistance to deformation of the pile–soil system increases; so, the vertical deformation in the pile–soil system will gradually decrease, which leads to a decrease in the maximum settlement value. Conversely, an increase in diameter enlarges the mass of the pile–soil system, leading to an increase in the compressive deformation of the soil layer beneath it, which leads to an increase in the maximum settlement value. This shows that in the deep burial pile–soil system, with the enlargement of the gravel pile diameter, the

increase in its lower fly ash compression deformation is first greater than and then less than the decrement of the vertical deformation of the pile–soil system.

5. Subdam-III was constructed in 2016; it has been standing for 7 years since then. Its deformation has stabilized. The settlement observation data of subdam-III are basically consistent with the settlement curve of subdam-III obtained in the simulation, which verifies the reasonableness of the parameters. As this part was not the main focus of this paper, it is not described in detail. The maximum settlement values of various scenarios in the predicted results range from 200 mm to 400 mm, and this also agrees well with the data observed in the operation and the management of other ash sites.

6. From the comprehensive perspective of the main deformation analysis and SD analysis, scheme 3 (with a center distance of 6 m and a diameter of 0.5 m) is therefore recommended as the actual construction scheme.

**Author Contributions:** Methodology, Y.T.; Investigation, X.L.; Data curation, L.H.; Writing—original draft, H.W.; Writing—review & editing, Z.Y.; Supervision, Y.W. All authors have read and agreed to the published version of the manuscript.

**Funding:** This research was funded by Sichuan Science and Technology Program (Grant No. 2023YFS0365).

**Informed Consent Statement:** Not applicable.

**Data Availability Statement:** No new data were created.

**Acknowledgments:** The authors would like to acknowledge Yong Wu for providing thesis supervision. We would like to thank Zongyao Yang of Chengdu Surveying Geotechnical Research Institute Co., Ltd. of MCC for offering suggestions for paper revision. The authors would like to express their gratitude to Xuefeng Li and Duoliang Yuan of SEPDI for their help and to thank Xiaotao Li, Shaowu Yang, and other managers of PPIG for cooperating in the field work. The authors would like to thank the State Key Laboratory of Geological Hazards and Environmental Protection for supporting the laboratory geotechnical experiment.

**Conflicts of Interest:** The authors declare no conflict of interest.

## List of Symbols

| | |
|---|---|
| $\sigma_1, \sigma_2, \sigma_3$ | Maximum, intermediate, and minimum principal stresses |
| $E$ | Modulus of elasticity |
| $c$ | Cohesion |
| $\sigma^t$ | Tensile strength |
| $\varphi$ | Friction angle |
| $f^s$ | Compression yield function |
| $f^t$ | Tensile yield function |
| $\alpha_1, \alpha_2$ | Material constants |
| $G$ | Shear modulus |
| $K$ | Bulk modulus |
| $V_i^t$ | Total volume of tension failure zones in $i$th scheme |
| $V_i^s$ | Total volume of shear failure zones in $i$th scheme |
| $V$ | Total volume of the model |
| $P_i^s$ | Percentage of shear failure zone |
| $P_i^t$ | Percentage of tension failure zone |
| $D_i^s$ | Reduction degree of shear failure zone of $i$th scheme |
| $D_i^t$ | Reduction degree of tension failure zone of $i$th scheme |

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
