# Peer review of "Numerical Simulation of Subdam Settlement in Ash Disposal Based on CGSW Optimization"

_applsci, doi:10.3390/app13148370_

Round 1
Reviewer 1 Report
- Please, highlight placement of dams in fig. 1.
- The literature survey of paper is the main concern. The past related studied in must be mentioned in the introduction section. what are the importance and novelty of the paper?
- Section 3.2 (lines 203-225) seems general and it is better to be summarized.
- It should be explained that why the finite-difference numerical method is used for study? All steps of modelling should be presented?
- Validation of the results needs to be clarified.
- The standard deviation analysis in lines 378-386 should be moved to section 3.
- Comparison of the subsidence curves to the real filed data should be explained.
- Quality of Fig. 9 needs to be improved.
Reviewer 2 Report
The authors studied numerically subdam settlement in ash disposal based on CGSW optimization. This manuscript is very complete and detalied. However, I have the following suggestions:
1. Page 8, Line 254 "We need to verify the feasibility of this idea. ". Maybe the authors would like to delete this sentence.
2. Page 13, Line 371 " When diameters are same, the maximum subsidence increases and then decreases with the increase of center distance." This seems a little confusing
Reviewer 3 Report
This manuscript studied the issue of “Numerical simulation of subdam settlement in ash disposal based on CGSW optimization”.
First of all, I would like to thank the authors of this manuscript for the effort they put into making it. The paper needs to be rewritten and its objectives well redefined. On the other hand, I have added some comments with the main objective of improving the manuscript. In my opinion, the subject of the paper is remarkably interesting. However, the paper needs some major revisions.
i. What is the innovation point or significance of the study for this article? Please make clear the novelty and contribution of the manuscript and its results as compared to the extensive literature available. The manuscript does not provide a clear objective of the study. What does it add to the subject area compared with other published manuscripts?
ii. Results are merely present and there are no scientific findings are discussed. What is the difference between this manuscript and the article, “Experimental Study on the Optimization of Coal-Based Solid Waste Filling Slurry Ratio Based on the Response Surface Method”?
iii. There is no consistency in labeling of the figures. “Fig” 6 in change to “Figure” 6 and etc.
iv. How did the author evaluate the soil void ratio from the model?
v. How are the parameters in Table 2 obtained?
vi. If the model is to simulate the realistic condition numerically, what is the effect of lowering the groundwater level? As the authors pointed out, the groundwater significantly affects the model. How does the author justify the groundwater level reduction in their model?
vii. Results are merely present and there are no proper results for satisfy the conclusions. The introduction on similar work is very limited and does not cover similar experiences on the topic. I strongly recommend authors give a broader overview of similar works on the topic. The introduction should focus on the content related to the topic of the article.
viii. All the parameters used in the text should be defined. I suggest the authors provide a nomenclature.
ix. In conclusions, the useful data is not provided and the words prove general and lack of academic contributions. Please provide more specific details, evidence and arguments in the conclusions. The current conclusion is rather generic. The discussion is rather basic and short.
x. DOIs are missing in the list of references.
The English language could be improved as well as the format of the manuscript; though overall English is acceptable.
Round 2
Reviewer 3 Report
The all comments are considered in the current revision. the article is proper for published.
The all comments are considered in the current revision.